# In Vitro Human Metabolism and Inhibition Potency of Verbascoside for CYP Enzymes

**DOI:** 10.3390/molecules24112191

**Published:** 2019-06-11

**Authors:** Anna-Mari Reid, Risto Juvonen, Pasi Huuskonen, Marko Lehtonen, Markku Pasanen, Namrita Lall

**Affiliations:** 1Department of Plant and Soil Sciences, University of Pretoria, Pretoria 0002, South Africa; annamarikok@gmail.com; 2School of Pharmacy, Faculty of Health Sciences, University of Eastern Finland, Kuopio FI-70210, Finland; risto.juvonen@uef.fi (R.J.); pasi.huuskonen@uef.fi (P.H.); markku.pasanen@uef.fi (M.P.); 3LC-MS Metabolomics Center, Biocentre, Kuopio, Kuopio FI-70210, Finland; marko.lehtonen@uef.fi; 4School of Natural Resources, University of Missouri, Columbia, MO 65211, USA; 5College of Pharmacy, JSS Academy of Higher Education and Research, Mysuru 570015, India

**Keywords:** cytochrome P450, verbascoside, metabolism, antioxidant, cytotoxicity, liver

## Abstract

Verbascoside is found in many medicinal plant families such as Verbenaceae. Important biological activities have been ascribed to verbascoside. Investigated in this study is the potential of verbascoside as an adjuvant during tuberculosis treatment. The present study reports on the in vitro metabolism in human hepatic microsomes and cytosol incubations as well as the presence and quantity of verbascoside within *Lippia scaberrima*. Additionally, studied are the inhibitory properties on human hepatic CYP enzymes together with antioxidant and cytotoxic properties. The results yielded no metabolites in the hydrolysis or cytochrome P450 (CYP) oxidation incubations. However, five different methylated conjugates of verbascoside could be found in S-adenosylmethionine incubation, three different sulphate conjugates with 3′-phosphoadenosine 5′-phosphosulfate (PAPS) incubation with human liver samples, and very low levels of glucuronide metabolites after incubation with recombinant human uridine 5’-diphospho-glucuronosyltransferase (UGT) 1A7, UGT1A8, and UGT1A10. Additionally, verbascoside showed weak inhibitory potency against CYP1A2 and CYP1B1 with IC_50_ values of 83 µM and 86 µM, respectively. Potent antioxidant and low cytotoxic potential were observed. Based on these data, verbascoside does not possess any clinically relevant CYP-mediated interaction potential, but it has effective biological activity. Therefore, verbascoside could be considered as a lead compound for further drug development and as an adjuvant during tuberculosis treatment.

## 1. Introduction

Verbascoside (Appendix A), also known as acteoside, is a phenylethanoid consisting of a cinnamic acid and hydroxyphenylethyl moieties attached to a β-glucopyranose through a glycosidic bond [1,2,3]. Verbascoside has extensively been studied for its in vitro and in vivo biological activities [1,4,5]. Nektarios et al. (2003) [6] investigated the free radical scavenging effect of verbascoside in vitro and found it was comparable to that of α-tocopherol. Verbascoside is also a known constituent to occur within the five *Lippia* species endemic to South Africa [7].

The kinetic properties and particularly the metabolism of a xenobiotic determines its internal dose and concentration in a specific target of the body. No extensive data are available on the pharmacokinetic properties of verbascoside. However, the pharmacokinetic properties of verbascoside, such as absorption and elimination rates, were previously studied in rats, and findings stated fast absorption and elimination [8]. Verbascoside has the ability to be absorbed into the blood stream and undergoes further metabolism [9,10]. A low oral bioavailability of verbascoside has been reported [8]. Cui et al. (2016) [10] investigated the metabolic profile of verbascoside in the presence of human and rat intestinal bacteria as well as with rat intestinal enzymes. This study provided an insight into the metabolic pathway of verbascoside and also established the fact that phenylethanoid glycosides were metabolized by both intestinal bacteria and enzymes, both in humans and rats. The low oral bioavailability in rats has been described to be due to the multiple routes of hydrolysis by the bacteria of the gastrointestinal ducts, with several degradation products as a result thereof. The structure of verbascoside suggests that it is most likely to be metabolized by hydrolyzing and conjugating enzymes, as it contains an ester and several hydroxyl groups.

In addition to being metabolized, verbascoside may cause herb–drug interactions through induction or inhibition of metabolic enzymes found in the liver or intestines [11]. The metabolic profiles of plant extracts and pure compounds are very important to determine if any future herb–drug metabolic interactions will occur. In the present study, ultra-high-performance liquid chromatography quadrupole time-of-flight mass spectrometry (UHPLC-QTOF-MS) was used to analyze metabolites of verbascoside after incubation with human liver microsomes or cytosolic proteins and in the presence of NADPH, UDP-glucuronic acid, S-adenosylmethionine, and 3′-phosphoadenosine 5′-phosphosulfate (PAPS). Also, the presence and quantity of verbascoside within *Lippia scaberrima*, an indigenous species of *Lippia* in South Africa, were investigated through UHPLC-QTOF-MS. This method served to confirm that the biological activity found for *Lippia scaberrima* in previous studies conducted may be attributed to the presence of verbascoside. Additionally, inhibition of hepatic CYP enzymes by verbascoside were studied with recombinant CYP enzymes, selective marker substrates of hepatic CYP enzymes, and placental CYP19A1. Also investigated were the antioxidant properties of verbascoside and the cytotoxic potential on both first (peripheral blood mononuclear cells—PBMC) and secondary (HepG2) cell lines.

## 2. Results

### 2.1. Identification of Verbascoside through Ultra-High-Performance Liquid Chromatography Quadrupole Time-Of-Flight Mass Spectrometry (UHPLC-QTOF-MS)

The presence and the calculated mass of verbascoside within *Lippia scaberrima* was investigated through UHPLC-QTOF-MS in negative ion mode. Identification of verbascoside was based on comparing retention times of pure standard and samples together with high-accuracy mass and isotopic pattern of the analyte and its metabolites. The content of verbascoside was 0.17 mg/mL or 6.8% of the total weight (Appendix A).

### 2.2. In Vitro Metabolism of Verbascoside

In vitro oxidation, hydrolysis, glucuronidation, sulfonation, and methylation metabolism of verbascoside was studied in the presence of human liver microsomes or cytosolic proteins and specific reactions requiring cofactors. The level of verbascoside decreased when it was incubated with UDP-glucuronic acid, PAPS, and S-adenosylmethionine, but no decrease was detected with hydrolysis and NADPH containing oxidation incubations (Table 1). S-adenosylmethionine incubation produced five different methylated conjugates of verbascoside, whose levels increased up to 60 min (Table 1). PAPS incubation produced three different sulphate conjugates of verbascoside (Table 1), whose levels increased up to 20 min. No glucuronide conjugate of verbascoside could be identified from the UDP-glucuronic acid incubations together with human liver microsomes. However, incubations with recombinant human UGT1A7, 1A10, and UGT1A8 produced low levels of glucuronide metabolites of verbascoside (Table 1)

### 2.3. Inhibition of CYP Activities by Verbascoside

To find out if verbascoside could inhibit individual human CYP enzymes, concentration-dependent inhibitions were determined for eight recombinant CYP enzymes and for selective CYP substrate assays with hepatic and placental microsomes (Table 2) (Appendix A). Verbascoside was a weak inhibitor for many CYP enzymes. The IC_50_ value of verbascoside was more than 50 µM for CYP1A2, 1B1, 2D6, and 3A4 and did not inhibit CYP1A1, 2A6, 2C19, and 19A1 (placental aromatase) enzymes.

### 2.4. 2,2-Diphenyl-1-Picrylhydrazyl (DPPH) Inhibitory Activity of Verbascoside

The 2,2-diphenyl-1-picrylhydrazyl (DPPH) radical scavenging assay is a method that indicates the free radical scavenging potential of samples or compounds [12]. During this investigation verbascoside showed potent DPPH radical scavenging activity with an IC_50_ value of 2.50 ± 0.02 µM, even lower than for the positive control, ascorbic acid, which had an IC_50_ value of 43.72 ± 1.12 µM (Table 3).

### 2.5. Nitric Oxide (NO) Inhibitory Activity of Verbascoside

According to Table 3, nitric oxide (NO) inhibitory activity of verbascoside was moderately effective when compared to that of the positive control, ascorbic acid, with IC_50_ values of 382.01 ± 4.15 µM and 143.94 ± 3.30 µM.

### 2.6. Cellular Antiproliferative Activity of Verbascoside

The antiproliferative activity of verbascoside was determined on both primary (PBMC) and secondary (HepG2) cell lines. Verbascoside had no toxic effects on both the PBMCs and the HepG2 cell lines, as both IC_50_ values obtained were above 100 µM (Table 3).

## 3. Discussion

Metabolism is an important component of the kinetic characteristics of herbal constituents, and it often determines internal dose and concentration of effective constituents at the target site. In this investigation we studied (1) the in vitro human hepatic metabolism of verbascoside, which is found from many medicinally used herbal plants, and (2) inhibition of hepatic and placental CYPs by verbascoside. Verbascoside was metabolized efficiently to methyl and sulphate conjugates, but it was not hydrolyzed or oxidized by human liver subcellular fractions. Three recombinant UGTs had the ability to conjugate verbascoside to glucuronides. Through this in vitro study it can be summarized that verbascoside, indeed, can undergo phase two metabolism in the liver and most probably also in the intestine. Therefore, it could explain the low oral bioavailability as seen in previous studies [13]. It was confirmed through UHPLC-QTOF-MS that verbascoside was present within the indigenous species of *Lippia scaberrima* and, therefore, may contribute to the biological activity found for the ethanolic plant extract.

The inhibition potency of verbascoside against human CYP enzymes was very weak and, therefore, indicated low potential of verbascoside for clinical herb–drug interactions. However, no nuclear receptor binding studies were carried out to resolve whether any receptor-mediated inductions of metabolizing genes could be detected. Instead, verbascoside had a weak ability to stimulate CYP2C19 activity, both with the recombinant CYP2C19 and within liver microsomes, which could be due to allosteric binding of the chemical on the metabolizing enzyme. However, all these are in vitro results, and additional confirmatory clinical trials are needed to confirm the clinical relevance of this finding. According to the current study, verbascoside was found to stimulate CYP2C19, indicating targeted specificity of the compounds, and no other activities were stimulated.

On the other hand, Lee et al. (2004) [14] found that verbascoside showed a protective effect after carbon tetrachloride-induced hepatotoxicity. One explanation could be that verbascoside could act as a “scavenger” against tetrachloride-induced toxicity. Verbascoside showed potent free radical scavenging activity compared to ascorbic acid (the positive control used). According to a study by Chen et al. (2012) [15], verbascoside had an IC_50_ value of 11.4 µM; a higher value was obtained during the current study. This may be due to discrepancies in the two methods used and the lab conditions, as DPPH was dissolved in ethanol and not methanol as indicated in the study done by Chen et al. (2012) [15]. The mechanism of action of free radical scavenging is believed to be due to a free hydroxyl group in the glucose moiety found within the structure of verbascoside [15]. Another study done by Koo et al. (2006) [16] also indicated the effect of verbascoside on DPPH radical scavenging. The EC_50_ value obtained by Koo et al. for verbascoside was 1.28 µM, which was fairly similar to the results found in the current study. Another study conducted by Koo et al. (2006) [16] showed that verbascoside had the ability to boost the endogenous antioxidative system. 

Nitric oxide inhibition is one of the main mechanisms of down-regulating inflammation during infection, especially on the protein level through iNOS (inducible nitric oxide synthase) inhibition. Verbascoside is known for its anti-inflammatory effects as seen through studies conducted both in vitro and in vivo [17,18,19]. During the current study, verbascoside had effective inhibition of NO when compared to the positive control, ascorbic acid, which had an IC_50_ value of 143.94 ± 3.30 µM. The main mechanism through which verbascoside exerts its inhibitory effect has been shown to be through the inhibition of AP-1 (activator protein-1) [20].

Verbascoside had no noteworthy levels of toxicity observed in the current study on both cell lines tested. A study conducted by Sipahi et al. (2016) [21] investigated the cellular toxicity of an aqueous extract of verbascoside on PBMCs. The associated IC_50_ value was found to be 384 µM. The difference in the values obtained may be due to the difference in the solvents that the extracts were dissolved in. In agreement with Etemad et al. (2015) [19], no cellular toxicity was detected for verbascoside on the HepG2 cell line after 24 and 72 h with IC_50_ values higher than the highest concentration tested of 400 µM. These values are comparable to the values obtained in the current study.

## 4. Materials and Methods

### 4.1. Cell Lines, Chemicals, and Reagents

Verbascoside with a purity of >99%, coumarin, 7-hydroxycoumarin, and finrozole were purchased from Sigma Aldrich (St. Louis, Mo, USA). Synthesis and purity of (TFD024 (3-(3-methoxyphenyl)-6-methoxycoumarin), OCA349(3-(4-trifluoromethylphenyl)-6-methoxycoumarin), TFD008_1 (3-(4-phenylacetate)-6-chlorocoumarin, coumarin, TFD032 (3-(3-methoxyphenyl)coumarin), TFD023 (3-(4-phenyl)-7-methoxycoumarin), and OCA369 (3-(3-benzyloxo)phenyl-7-methoxycoumarin) were described in published papers [22,23,24]. Tris-HCl, magnesium chloride (MgCl_2_), MnCl_2_, isocitric acid, isocitric acid dehydrogenase, Glycin, NaOH, and trichloroacetic acid (TCA) were all bought from Sigma-Aldrich (Steineim, Germany). Androstenedione and ^3^H-Androst-4-ene,3,17-dione were obtained from PerkinElmer. KCl was from J.T. Baker (Loughborough, UK) and nicotinamide adenine dinucleotide phosphate (NADP^+^) was bought from Roche Diagnostics (Mannheim, Germany). The NADPH regenerative system consisted of 1.12 mM NADP^+^, 12.5 mM MgCl_2_, 12.5 MnCl_2_, 16.8 mM isocitric acid, 0.056 mM KCl, and 15 U isocitric acid dehydrogenase in 188 mM Tris-HCl buffer at pH 7.4. cDNA-expressed human wild-type CYPs (CYP1A1, CYP1A2, CYP2A6, CYP3A4, CYP1B1, CYP2C19, CYP2D6, and CYP19A1) were obtained from BD Biosciences Discovery Labware (Bedford, MA, USA). Livers used during this study were obtained from University of Oulu Hospital as excess from kidney transplantation donors. The excess tissue collection was approved by the Ethics committee of the Medical Faculty of the University of Oulu (Doc 01-38; 1 June 2000). Liver samples were surgically excised and immediately transferred to ice, then they were cut into pieces, snap frozen in liquid nitrogen, and stored at −80 °C until microsomal preparation. Liver microsomes were prepared as described by Lang et al. (1981) [25]. Microsomal protein concentration was determined by using the Bradford method (1976) [26]. Placental microsomes were obtained from both nonsmoking and smoking mothers of previous studies and prepared according to Huuskonen et al. (2015) [27]. The placental microsomes as part of the KuBiCo plan were approved by The Research Ethics Committee of the hospital district of Central Finland in Jyväskylä, Finland 15.11.2011. The HepG2 cell line (HB-8065) was obtained from American Type Culture Collection (ATCC). Cell culture materials and reagents such as fetal bovine serum (FBS), Dulbecco’s modified Eagle’s medium (DMEM) media, trypsin-EDTA, and antibiotics were supplied by Highveld Biological (Pty) Ltd. (Johannesburg, RSA). Ficoll-Hypaque was obtained from Pharmacia Biotech (Piscataway, NJ, USA). PrestoBlue was purchased from Thermo Fisher (Carlsbad, CA, USA). All reagents obtained were of analytical grade. The XTT Cell Proliferation Kit II was supplied by Roche Diagnostics (Pty) Ltd. (Johannesburg, RSA). ACK (ammonium, chloride, and potassium) was obtained from Life Technologies (Carlsbad, CA, USA). Trypan blue solution was supplied from Thermo Fischer Scientific (Waltman, MA, USA). All chemicals were of the highest purity available from the suppliers.

### 4.2. In Vitro Metabolism and Identification of Verbascoside through UHPLC-QTOF-MS

To determine in vitro metabolism, 10 µM verbascoside was incubated for 1 h at 37 °C in 1 mL of 100 mM phosphate buffer (pH 7.4) with 400 µg human hepatic microsomal or cytosolic protein with different cofactors. From the incubation mixture, 450 µL acetonitrile was added to 150 µL of the samples to stop the reaction at time points of 0, 10, 20, 40, and 60 min. The CYP incubation mixture contained 20% NADPH regenerating system, whereas the glucuronidation incubation mixture contained 0.5 mM UDP-glucuronic acid with 5 mM MgCl_2_. The sulfonation incubation mixture contained 10 µM PAPS, 5 µM MgCl_2_, and 500 µg cytosolic protein instead of microsomes, and the methylation incubation mixture contained 0.5 mM S-adenosylmethionine, 5 mM MgCl_2_, and 500 µg cytosolic protein instead of microsomes. Blank samples did not contain any of the cofactors. The hydrolysis reaction was studied with an incubation mixture that contained 400 µg microsomal protein, or 500 µg cytosolic proteins, and 5 mM MgCl_2_. A sample with the absence of any enzyme was considered as the blank sample. After stopping the reactions, the samples were centrifuged for 20 min at 10,000× *g* and stored at −80 °C for analysis by the UHPLC-QTOF-MS system (Agilent Technologies, Waldbronn, Karlsruhe, Germany), which consisted of a 1290 LC system, a Jetstream ESI source, and a 6540 UHD accurate mass quadrupole time-of-flight (QTOF) mass spectrometry. The sample tray was kept at 4 °C during these analyses. UHPLC-QTOF-MS was used to both identify the metabolites formed after metabolism of verbascoside and the quantity of verbascoside found within the ethanolic extract of *Lippia scaberrima* (Appendix A). Two microliters of sample were injected onto a column (Zorbax Eclipse XDB-C8, 2.1 × 100mm, 1.8 µm, Agilent Technologies, Palo Alto, CA, USA) that was kept at 50 °C. Mobile phases, delivered at 0.4 mL/min, consisted of water (eluent A) and methanol (eluent B), both containing 0.1% (*v*/*v*) of formic acid. The gradient employed was as follows: 2%→100% B (0–10 min); 100% B (10–14.5 min); 100%→2% B (14.5–14.51 min); and 2% B (14.51–16.50 min).

A Jetstream ESI source, operated in negative ionization mode, used the following conditions: drying gas temperature 325 °C and a flow of 10 L/min, sheath gas temperature 350 °C and a flow of 11 L/min, nebulizer pressure 45 psi, capillary voltage 3500 V, nozzle voltage 1000 V, fragmentary voltage 100 V, and skimmer 45 V. Nitrogen was used as the instrument gas. For data acquisition, a 2 GHz extended dynamic range mode was used from m/z 20 to m/z 1600. Data were collected in the centroid mode at an acquisition rate of 1.67 spectra/s with an abundance threshold of 150. The TOF was calibrated on a daily basis and subsequently operated at a high accuracy (<2 ppm). Continuous mass axis calibration was performed by monitoring two reference ions from an infusion solution throughout the runs. The reference ions were m/z 112.985587 and m/z 966.000725. Identification of the metabolites found in the treatments was based on accurate mass and isotope information (i.e., ratios, abundances, and spacing). The software for the identification of metabolites used was MassHunter Metabolite ID B.04.00 (Agilent Technologies, Santa Clara, CA, USA).

### 4.3. Inhibition of Placental (CYP1A1), Microsomal, and Recombinant (CYP1A2, CYP2A6, CYP3A4, CYP1B1, CYP2C19, and CYP2D6) Oxidation by Verbascoside

IC_50_ determinations of verbascoside for human recombinant CYPs, human liver microsomes, and placental microsomes were determined by the methods described by Juvonen et al. (2018) [24], Huuskonen et al. (2016) [28], and Crespi et al. (1997) [29]. Incubations were carried out in a total volume of 100 µl of 100 mM Tris-HCl buffer (pH 7.4) in all black, flat-bottom Costar 96-well plates (Corning Incorporated, Corning, NY, USA). The reaction mixtures contained a 20% NADPH-regenerating system in Tris-HCl buffer (pH7.4), 10 µM coumarin or its derivative substrate ((TFD024 (3-(3-methoxyphenyl)-6-methoxycoumarin), OCA349 (3-(4-trifluoromethylphenyl)-6-methoxycoumarin), TFD008_1 (3-(4-phenylacetate)-6-chlorocoumarin, coumarin, TFD032 (3-(3-methoxyphenyl)coumarin), TFD023 (3-(4-phenyl)-7-methoxycoumarin) or OCA369 (3-(3-benzyloxo)phenyl-7-methoxycoumarin), 0.010 g/L microsomal protein or 5 nM recombinant CYP, and inhibiting agent of 0.4, 2, 10, and 50 µM verbascoside. A test control (100%) did not contain any inhibiting agent, and blank reactions did not contain any substrate or enzyme. Verbascoside was added from a stock solution containing ethanol so that the final concentration of ethanol in the incubation mixture was at 1%. The samples were then preincubated at 37 °C for 10 min. The reactions were initiated by the addition of NADPH, and 0–10 µM 7-hydroxycoumarin was used as the standard. The fluorescent signal was measured with a Victor^2^ 1420 multilabel plate counter (PerkinElmer Lifesciences Wallac, Turku, Finland) with excitation and emission wavelengths of 405 and 460 nm, respectively. Fluorescence was monitored every 2 min for 40 min from where the concentration was calculated at the various time points. From these, oxidation rates (µM/min) and the relative remaining activity of the sample at different concentrations were calculated. The data were fit to sigmoidal dose-response curves with nonlinear regression, and IC_50_ values (the concentration at which the sample reduced the metabolism of the CYP substrate by 50%) were determined. The calculation was based on the equation vi/v0 = 1/(1 + i/IC_50_), in which (vi) is the rate at the specific concentration of the inhibiting agent, (v0) is the rate without the inhibitor, (IC_50_) is the sample concentration with 50% inhibition, and (i) is the inhibitor concentration. This was analyzed using nonlinear equations in Graph Pad Prism, which produced IC_50_ values and its 95% confidence interval as the results. Degree of inhibition was categorized as potent (IC_50_ < 1 µM), marginal/moderate (1 µM < IC_50_ > 10 µM), weak (IC_50_ > 10 µM), or no inhibition (IC_50_ > 100 µM) [30].

### 4.4. Inhibition of Placental CYP19A1 Oxidation

To determine whether verbascoside inhibited CYP 19A1 (aromatase), an aromatase inhibition assay was performed according to the method of Pasanen (1985) [31]. The relative remaining aromatase activity was calculated by measuring the tritiated water (^3^H_2_O) formed from the aromatization of ^3^H-Androst-4-ene,3,17-dione. Each of the incubation tubes contained 50 µM ^3^H-Androst-4-ene,3,17-dione; androstenedione in an acetone–tween solution, distilled water; 100 mM Tris-HCl buffer at pH 7.4; placental microsomes; either 4, 20, 100, or 500 µM verbascoside; 1% DMSO; and 1 or 10 µM Finrozole as a positive control. Twenty percent NADPH regenerative system was added to initiate the reaction. Reactions were terminated by the addition of 33% trichloroacetic acid. The supernatant was investigated for tritiated water formed in a scintillation mixture (Optiphase Hisafe 2, Perkin Elmer) using a Wallac microbeta 1450 TriLux liquid scintillation and luminescence counter (PerkinElmer). The degree of inhibition was categorized as potent (IC_50_ < 1 µM), marginal/moderate (1 µM < IC_50_ > 10 µM), weak (IC_50_ > 10 µM), or no inhibition (IC_50_ > 100 µM) [30].

### 4.5. DPPH Inhibitory Activity of Verbascoside

The antioxidant properties of verbascoside were determined by the method described by Berrington and Lall (2012) [32]. Stock solutions of verbascoside and ascorbic acid (positive control) were prepared at 10 mg/mL and 2 mg/mL in 100% ethanol, respectively. Twenty microliters of verbascoside stock solution was added to the top well of a 96-well plate and serially diluted to concentrations that ranged from 3.90–500 µg/mL. Ascorbic acid was serially diluted to a concentration range of 0.78–100 µg/mL. Ethanol (10%) was used as a blank. Ninety microliters of a 0.04 M DPPH (2,2-diphenyl-1-picrylhydrazyl) in ethanol were added to each well. The plates were incubated for 30 min and covered in a layer of foil. Color controls (negative controls) were prepared in the exact same manner as above, but distilled water was added to each well instead of DPPH. The absorbances were measured at a wavelength of 515 nm using a BIO-TEK Power-Wave XS multiplate reader (BIO-TEK Instruments, Winooski, VT, USA). From the absorbances an IC_50_ value was determined. All concentrations of verbascoside and ascorbic acid were tested in triplicate. 

### 4.6. NO Inhibitory Activity of Verbascoside

The nitric oxide scavenging properties of verbascoside were determined according to the method of Twilley et al. (2017) [33]. Verbascoside and the positive control (ascorbic acid) were all prepared in ethanol to stock concentrations of 10 mg/mL. Verbascoside and ascorbic acid (20 µL) were added to the top wells of a 96-well plate. Serial dilutions were made, and the concentration range for verbascoside and ascorbic acid ranged between 12.50–1601.05 µM and 44.35–5677.95 µM, respectively. Ethanol (10%) was used as a blank. Nitroprusside (50 µL) at a concentration of 10 mM was added to all the wells. The plates were incubated for 90 min at room temperature. After incubation, Griess reagent (100 µL) was added to all the wells, except for the color controls (negative controls) where instead distilled water was added. The absorbances were read at a wavelength of 546 nm using a BIO-TEK Power-Wave XS multiplate reader. From the absorbances an IC_50_ value was determined. All concentrations of verbascoside and ascorbic acid were tested in triplicate.

### 4.7. Cellular Antiproliferative Activity of Verbascoside

To determine the cellular viability of verbascoside against both first (PBMC) and secondary (HepG2) cell lines, the method of Berrington and Lall. (2012) [31] was performed. PBMC isolation and testing was approved by the Ethics Committee of the Department of Natural and Agricultural Sciences, University of Pretoria, Pretoria (Doc nr: EC120411-046; 17.05.2012). Briefly, peripheral blood mononuclear cells were isolated from blood that was freshly donated on the same day at the Student Clinic at the University of Pretoria. Mononuclear cells were isolated by centrifugation over Ficoll-Hypaque, a density gradient solution, with a density gradient of 1.07 g/mL. A layer of fresh heparinized venous blood was layered on the Ficoll-Hypaque at a ratio of 1:1 with supplemented RPMI 1640 media and then subjected to centrifugation at 3000 rpm for 30 min. Through centrifugation, the whole blood sample was separated into its different layers. As observed from the top, the different layers consisted of the plasma with its other constituents followed by a white buffy-coated layer of mononuclear cells (the PBMCs). After the PBMC layer, a layer of Ficoll was found and then finally a layer of erythrocytes (red blood cells) and granulocytes. The white, buffy-coated layer of PBMCs was aspirated out gently and transferred aseptically to a sterile Falcon tube (50 mL). The resultant suspension of cells was then washed with supplement RPMI 1640 media (30 mL) with an antibiotic (Gentamycin 1%) and subjected to centrifugation at 2200 rpm for 10 min. The supernatant was aspirated from the tube, and the resultant pellet of cells was washed with ACK (ammonium, chloride, and potassium). ACK is a lysis buffer that causes any contaminant red blood cells in the solution to be lysed. After 5 min with the added ACK, supplemented RPMI 1640 media with Gentamycin was added and subjected to centrifugation at 1200 rpm for 10 min. The supernatant was poured off, and the resultant pellets of cells were resuspended in approximately 5 mL of supplemented RPMI 1640 media with Gentamycin. The cells were counted and seeded in 96-well plates with a cell density of 10,000 cells/mL. The following formula was used to adjust the cell concentrations (Equation 1): (1)$$\mathrm{Concentration}\;\mathrm{of}\;\mathrm{the}\;\mathrm{cells}=\frac{\mathrm{Cells}\;\mathrm{counted}}{\mathrm{number}\;\mathrm{of}\;\mathrm{blocks}\;\mathrm{counted}}\times 10000.$$

The HepG2 cell line was seeded with a concentration of 1 × 10^5^ cells/well. The plates were incubated for 24 h at 37 °C and 5% CO_2_ to allow for cellular attachment to the bottom of the wells. Verbascoside was prepared to a stock solution of 2000 μg/mL. Serial dilutions were made with final test concentrations ranging from 1.53–400 μg/mL. Plates were incubated for 48 h at 37 °C and 5% CO_2_. DMSO as a solvent control (DMSO 2%) and Actinomycin D as a positive control, with a final test concentration of 0.002–0.5 μg/mL, were added. After incubation, PrestoBlue (20 μL) was added to each well, and the plates were further incubated for 4 h. The absorbance was read at 490 nm with a reference wavelength of 690 nm using a BIO-TEK Power-Wave XS multiwell plate reader. The assay was performed in triplicate, and the mean IC_50_ values were calculated. 

## 5. Statistical Analysis

Statistical analysis was performed with GraphPad Prism (Version 7, Graphpad Software, San Diego, CA, USA) using one-way analysis of variance (ANOVA). The results are expressed as the mean ± standard deviation, and *n* = 3 or more.

## 6. Conclusions

In the current study verbascoside was found to undergo extensive conjugation metabolism and had no significant inhibitory potential against the most important CYPs found to metabolize many known drugs and compounds. Verbascoside was also found to have noteworthy antioxidant and anti-inflammatory potential as well as low levels of cellular toxicity. All these activities indicate the potential of verbascoside to act as an adjuvant compound that can be used during first-line drug treatment of tuberculosis, as it holds no potential of herb–drug interactions and possesses important additional biological activity.

## Figures and Tables

**Table 1 molecules-24-02191-t001:** In vitro metabolites of verbascoside (VMs are methylation vs. sulfate and VGs are glucuronide conjugates of verbascoside). Metabolites are identified from duplicates of time-dependent experiments described in detail in the Section 4.

Metabolite	RT (min)	Calculated Mass	Formula	*m*/*z*	Δ Mass (ppm)	Score of Isotopic Pattern Matching
Verbascoside	4.762	624.2054	C_29_H_36_O_15_	623.1989	1.17	99.6
Methyl conjugation						
VM1	5.25	638.2211	C_30_H_38_O_15_	637.2141	0.40	96.1
VM2	5.336	638.2211	C_30_H_38_O_15_	637.2137	−0.40	98.7
VM3	5.438	638.2211	C_30_H_38_O_15_	637.2130	−1.39	84.9
VM4	5.603	638.2211	C_30_H_38_O_15_	637.2134	−0.68	95.2
VM5	5.734	638.2211	C_30_H_38_O_15_	637.2132	−0.93	94.2
Sulfonation conjugation						
VS1	4.516	704.1622	C_29_H_36_O_18_S	703.1557	1.08	99.0
VS2	4.773	704.1622	C_29_H_36_O_18_S	703.1561	1.64	66.9
VS3	4.976	704.1622	C_29_H_36_O_18_S	703.1551	0.25	85.2
Glucuronide conjugation						
VG1	3.96	800.2375	C_35_H_43_O_21_	799.2306	0.51	94.5
VG2	4.49	800.2375	C_35_H_43_O_21_	799.2313	1.27	91.5
VG3	4.76	800.2375	C_35_H_43_O_21_	799.2313	1.43	99.0

**Table 2 molecules-24-02191-t002:** Inhibitory concentrations (µM) of verbascoside against recombinant CYP and hepatic or placental microsome-catalyzed CYP oxidations. IC_50_ was calculated from verbascoside concentration-dependent experiments of duplicate samples as described in detail in the Section 4.

		Recombinant CYP	Microsomal
CYP Enzyme	Substrates	IC_50_ (µM) with 95% confidence intervals	IC_50_ (µM) with 95% Confidence Intervals
CYP 1A1	TFD024	N/A ^a^	350 (0–870)
CYP 1A2	OCA349	83 (21–144)	24 (0–48)
CYP 1B1	TFD008_1	86 (71–101)	No inhibition ^b^
CYP 2A6	coumarin	No inhibition	135 (92–180)
CYP 2C19	TFD032	Stimulation ^c^	
CYP 2D6	TFD023	131 (37–225)	96 (21–171)
CYP 3A4	OCA369	314 (0–953)	76 (24–129)
CYP 19A1	Androstendione ^d^	N/A	No inhibition

^a^ N/A: No recombinant CYP1A1 available, CYP1A1 is only available within placental microsomes of tobacco smoking mothers; ^b^ No inhibition of CYP enzyme at the various concentrations tested; ^c^ Stimulation of CYP enzyme means the increase of activity at the various concentrations tested. IC_50_ values (µM) (the concentration at which the sample reduced the metabolism of the CYP substrate by 50%) were calculated for recombinant CYP enzymes, placental, and liver microsomes. The IC_50_ values and the 95% confidence intervals (CI) were determined from the appropriate dose-response curves. ^d^ No inhibition of CYP19A1 was observed for verbascoside. As a positive control for the assay, finrozole (1 and 10 μM) was used within human placental microsomes.

**Table 3 molecules-24-02191-t003:** The antioxidant (2,2-diphenyl-1-picrylhydrazyl (DPPH) and nitric oxide (NO)) and cellular antiproliferative properties of verbascoside.

Sample	DPPH IC_50_ ^a^ (µM)	NO IC_50_ (µM)	Peripheral Blood Mononuclear Cell (PBMC) IC_50_ (µM)	HepG2 IC_50_ (µM)
Verbascoside	2.50 ± 0.02	382.01 ± 4.15	169.55 ± 3.73	>640.42
Ascorbic acid ^b^	43.72 ± 1.12	143.94 ± 3.30	-^c^	-

^a^ Inhibitory concentration (50%); ^b^ Positive control for both DPPH and NO inhibitory assays; ^c^ Not applicable.

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
