# Peer review of "In Vitro Human Metabolism and Inhibition Potency of Verbascoside for CYP Enzymes"

_molecules, 2019, doi:10.3390/molecules24112191_

Round 1
Reviewer 1 Report
The authors explore the in vitro human metabolism and inhibition potency of 3-verbascoside for CYP enzymes, a study design of fundamental importance in the context of TB. The manuscript is well written with a thorough description of the methodological approaches. The rationale of the study is clear. Data are well presented and support the claims made. This reviewer's comments are presented below:
The authors should clarify if verbascoside is extracted from Lippia scaberrima and then tested in vitro plus its extraction parameters of importance (recovery rate, purity, quantity). The authors need to clarify per experimental set up if pure verbascoside has been tested. If so, please comment on its synthesis or extraction as well as quantification. Eg l.207 (and elsewhere).
The authors need to clarify their technical and biological replicates.
Do the authors feel they have encountered any cell line specific events? (cellular antiproliferative activity). Please comment.
In terms of statistical analyses, the authors need to clarify the test-groups in question (which was compared with what and how)
Author Response
The authors explore the in vitro human metabolism and inhibition potency of 3-verbascoside for CYP enzymes, a study design of fundamental importance in the context of TB. The manuscript is well written with a thorough description of the methodological approaches. The rationale of the study is clear. Data are well presented and support the claims made. This reviewer's comments are presented below:
The authors should clarify if verbascoside is extracted from Lippia scaberrima and then tested in vitro plus its extraction parameters of importance (recovery rate, purity, quantity). The authors need to clarify per experimental set up if pure verbascoside has been tested. If so, please comment on its synthesis or extraction as well as quantification. Eg l.207 (and elsewhere).
Answer: Verbascoside was not extracted, but testing was done to confirm the presence and quantity of verbascoside within Lippia scaberrima. The verbascoside used during the studies was bought from Sigma Aldrich, as indicated in Materials and methods
The authors need to clarify their technical and biological replicates.
Answer: Description of replications was added to the tables.
Do the authors feel they have encountered any cell line specific events? (cellular antiproliferative activity). Please comment.
Answer: No, HepG2 cell line is most often used for studies of drug metabolism and toxicity, whereas PBMCs are used during immunomodulatory studies
In terms of statistical analyses, the authors need to clarify the test-groups in question (which was compared with what and how)
Answer: 95 % confidence intervals are given for IC50-values. They were obtained from the calculated results of data tables. This was added to materials and methods.

Reviewer 2 Report
Introduction
Line 35: Provide references for the statement
Line 39: The word kinetics in the sentence should be replaced with metabolism which encompasses the statement authors want to put across. Xenobiotic disposition is not only affected by kinetics alone.
Line 57: The most appropriate term would be cytosolic proteins
Results
General comments: I couldn't access the supplementary materials
Results
Line 75: cytosolic proteins
Table 1: Define calculated mass. The data was collected in the negative ionazation mode which means the elemental composition will lack one H. Modify table to reflect the Molar mass of the negative ionization mode and also provide MS/MS fragmentation which aided in identification of the various compounds. What was the cut off point for the ppm?
Table 2:
Can authors provide the graphs plotted for the IC50 determination? What control compounds were used for each of the CYPs analyzed? Authors defines the IC50 inhibition with ranges however, there are values that are exceeding 100uM. What values were used in plotting the graphs and having figures of the graphs will be very helpful in understanding Table 2
Table 3: Does DPPH used the term IC50 or AA% to determine the scavenging activity of the compound?
Antiproliferative activity should rather have IC50 as a measuring tool (HepG2 data)
Discussion
Line 126: The use of first pass metabolism is not right? Phase one, phase II should be the terms used
Line 135: Delete the word "the"
Line 137-140: Authors did not provide any data for CYP2C19 from the results provided.
Line 160-162: Based on this Don't authors think that the solvent used could have had an effect on the results obtained. This should have been worked out in the methodology section with the right controls. This makes the comparison of the data obtained here inaccurate?
Line 163-167: What is the solvent effect in your experiment? Appropriate controls?
Line 168-170:Table 3 only shows results for 48-hours.
Materials and Methods
General comments: Authors have combined a lot of methods which need to divided into sections to make it easy to read.
Authors also add data analysis to materials and methods. To make the paper succinct, authors should consider having a section on data analysis for the various analysis performed.
Line181: City, Country for JT baker
Line 189: is Doc 01-38; June-1; 2000). the ethics number provided? Can authors provide the ethics number provided by the ethics committee
Line 192-193: Provide reference for Bradford method.
Line 196: delete The word "The"
Line 206: To stop the reaction, is ACN not added to the reaction? The methods mentioned says that reaction was added to the ACN. Kindly explain
Looking the volume of ACN added, what will be the effect on the final outcome of the compound concentration (verbascoside)
Materials and Methods
Line 227: What type of elution was employed? Gradient, isocratic. two eluents were stated but but authors only mention eluent b through out.
4.3: Inhibition of placental......
This section needs to be divided into sub-sections as it is very confusing moving from one method to the other.
Line 246: Why black 96-well plates?
Line 254: replace full reaction with test control. Blanks are suppose to contain all components of test control except substrates which is replaced with water or another solvent.
Line 255: What substances are authors referring to here?
Line 255-257: This statement is confusing. What was at 1%. Solvent or compound? if compounds were dissolved in EtOH, then there should be a solvent control as blank to take out the effects of EtOH on the reaction. Can authors address this issue?
Line 261: Per the statement it is understood that there are about 20 data points for calculating remaining activity and subsequent IC50 plotting?
Line 263: Data is the same for singular and plural so delete were and replace with was
Line 268: This sentence is confusing. There is a word missing
Line 270: what relative remaining activity was performed here? If this is a different method, authors should sectionalized the various methods to make the methodology readable as in its current state it is confusing
4.4 DPPH and NO......
Break this into two sections
What was the final concentration of EtOH in the reaction as this has effects on the DPPH assay. How was the 10% EtOH concentration chosen as blank?
Line 289: Define colour or negative controls
Line 290: Check the word Absorbencies????
Line 293: Check the word Absorbencies??
Line 299: Same as above
Line 302: same as above
Line 331: Sentence is wrong. Kindly rephrase
What solvent was used in this session?
Concentrations should be from 1.53-400 not 400- to 1.53ug/mL. range should be from low to high unless units are not the same
Same as above
Author Response
Line 35: Provide references for the statement
Answer: Added three references to substantiate statement made. See references 1, 4 and 5 in reference list.
Line 39: The word kinetics in the sentence should be replaced with metabolism which encompasses the statement authors want to put across. Xenobiotic disposition is not only affected by kinetics alone.
Answer: This change had been made accordingly- see comments made
Line 57: The most appropriate term would be cytosolic proteins
Answer: This change had been made accordingly- see comments made
Results
General comments: I couldn't access the supplementary materials
Results
Line 75: cytosolic proteins
Answer: This change had been made accordingly- see comments made
Table 1: Define calculated mass. The data was collected in the negative ionazation mode which means the elemental composition will lack one H. Modify table to reflect the Molar mass of the negative ionization mode and also provide MS/MS fragmentation which aided in identification of the various compounds. What was the cut off point for the ppm?
Answer: Calculate mass is the theoretical monoisotopic mass of the molecule (i.e., verbascoside and its metabolites). The term could be changed to monoisotopic mass in the table.
Molar mass of the negative ion is presented in table in section m/z. Delta mass is presenting the bias (or error of the mass) between measured mass and theoretical monoisotopic mass. The instrument´s mass accuracy is lower than 2 ppm.
We identify the substrate and metabolites with high resolution mass spectrometer with information of retention time (for substrate), accurate mass and isotopic pattern. In addition, each experiment contain blank sample where metabolites were not present. Further, the information that in the enzyme reaction conditions only certain types of metabolized could be formed (methylated, sulfonated or glucuronidated conjugates of verbascoside).
We consider this information adequate for positive identification of verbascoside and its metabolites.
We did measure product ion spectrums (MS/MS) from verbascoside and its metabolites. We were unable to obtain any MS/MS spectrum to glucuronide metabolites due to their low signal levels. The MSMS data was collected with data dependent matter with following parameter: For automatic data dependent MS/MS analyses, the precursor isolation width was 1.3 Da, and from every precursor a scan cycle of 4 most abundant ions were selected for fragmentation. These ions were excluded after two product ion spectra and released again for fragmentation after a 0.25 min hold. Precursor scan time was based on ion intensity, ending at 25,000 counts or after 300 ms. Product ion scan time was 300 ms. Collision energies were 10, 20, and 40 V in subsequent runs.
Table 2:
Can authors provide the graphs plotted for the IC50 determination? What control compounds were used for each of the CYPs analyzed? Authors defines the IC50 inhibition with ranges however, there are values that are exceeding 100uM. What values were used in plotting the graphs and having figures of the graphs will be very helpful in understanding Table 2
Answer: IC50 graphs supplied in the supplementary material. The program can calculate IC50-value although the inhibition do not reach high inhibition level. Our calculations are based on several inhibition concentrations. 95 % confidence limit is lower, when the experimental values are close to theoretical values and when the experimental values differ much from the theoretical values, 95 % confidence value is big. (Equation vi/v0 = 1/( 1 + IC50/I can be transformed to the form IC50 = (I * vi/v0) / ( 1 – vi/v0), which indicate that IC50-value can be estimated even with one inhibition concentration )
Table 3: Does DPPH used the term IC50 or AA% to determine the scavenging activity of the compound?
Answer: IC50- Fifty percent inhibitory concentration is the concentration of the sample that is required to cause 50% decrease in DPPH activity
Antiproliferative activity should rather have IC50 as a measuring tool (HepG2 data)
Answer: We agree that antiproliferative activity could be used as an alternative marker. However, as the main focus was in metabolism, we preferred IC50 value as the measuring tool
Discussion
Line 126: The use of first pass metabolism is not right? Phase one, phase II should be the terms used
Answer: This change had been made accordingly- see comments made
Line 135: Delete the word "the"
Answer: This change had been made accordingly
Line 137-140: Authors did not provide any data for CYP2C19 from the results provided.
Answer : We could not give the value calculated as they do not fall within the concentration ranges that we investigated. Their confidence intervals will be large.
Line 160-162: Based on this Don't authors think that the solvent used could have had an effect on the results obtained. This should have been worked out in the methodology section with the right controls. This makes the comparison of the data obtained here inaccurate?
Answer: Sentence removed but our solvent - ethanol - showed no significant inhibition as showed by verbascoside dissolved in PBS which had a lower IC50 concentration, indicating lower concentrations needed to obtain 50% inhibition. An ethanol solvent control was used during the current study to indicate any effect that may be due to the ethanol (solvent) none were indicated during the studies (results not shown)
Line 163-167: What is the solvent effect in your experiment? Appropriate controls?
Answer: Different solvents can be used to prepare your herbal extract- in this cause an ethanolic extract was made and tested accordingly in the studies. Different extracts will extract different compounds from the plant sample due to differences in polarities between compounds within the extract for example ethanol extracts compounds which are both polar and non-polar due to the structure of ethanol. Ethanol is also safe for human consumption which will support further studies into the ethanolic extracts of herbal extracts and in this case studies in to the ethanolic dissolved compound of verbascoside. Ethanol was used throughout my studies.
Line 168-170:Table 3 only shows results for 48-hours.
Answer: This change had been made accordingly
Materials and Methods
General comments: Authors have combined a lot of methods which need to divided into sections to make it easy to read.
Answer: Sections of results and materials and methods have been divided to subsections in the manuscript.
Authors also add data analysis to materials and methods. To make the paper succinct, authors should consider having a section on data analysis for the various analysis performed.
Answer: This change had been made accordingly
Line181: City, Country for JT baker
Answer: The city and country inserted.
Line 189: is Doc 01-38; June-1; 2000). the ethics number provided? Can authors provide the ethics number provided by the ethics committee
Answer: This information details the ethics permission.
Line 192-193: Provide reference for Bradford method.
Answer: Reference has been provided.
Line 196: delete The word "The"
Answer: This change had been made accordingly
Line 206: To stop the reaction, is ACN not added to the reaction? The methods mentioned says that reaction was added to the ACN. Kindly explain
Answer: This change had been made accordingly. Acetonitrile was added to the sample to stop the reaction
Looking the volume of ACN added, what will be the effect on the final outcome of the compound concentration (verbascoside)
Answer: Volume changes have been taken into consideration when calculating final results throughout all experiments. High concentrations of ACN precipitates better proteins.
Materials and Methods
Line 227: What type of elution was employed? Gradient, isocratic. two eluents were stated but but authors only mention eluent b through out.
Answer: The gradient employed was as follows: 2% →100% B (0-10 min); 100% B (10-14.5 min); 100%→2% B (14.5-14.51 min); 2% B (14.51-16.50 min). The elution conditions are described in the subsection of 4.2. In vitro metabolism and identification of verbascoside through UHPLC-QTOF-MS of Materials and methods.
4.3: Inhibition of placental......
This section needs to be divided into sub-sections as it is very confusing moving from one method to the other.
Answer: These changes have been made accordingly
Line 246: Why black 96-well plates?
Answer: To reduce the amount of background noise due to the reflection of light which will be higher when using white plates instead of black plates
Line 254: replace full reaction with test control. Blanks are supposed to contain all components of test control except substrates which is replaced with water or another solvent.
Answer: This change had been made accordingly- see comments made
Line 255: What substances are authors referring to here?
Answer: This change had been made accordingly- see comments made. Substances are indeed referring to the sample which is Verbascoside
Line 255-257: This statement is confusing. What was at 1%. Solvent or compound? if compounds were dissolved in EtOH, then there should be a solvent control as blank to take out the effects of EtOH on the reaction. Can authors address this issue?
Answer: 1% is referring to the concentration of EtOH
Line 261: Per the statement it is understood that there are about 20 data points for calculating remaining activity and subsequent IC50 plotting?
Answer: That is indeed correct- each concentration tested was read every 2 minutes for a time period of 20 min
Line 263: Data is the same for singular and plural so delete were and replace with was
Answer: This change had been made accordingly- see comments made
Line 268: This sentence is confusing. There is a word missing
Answer: I could not find the missing word. I have changed a sample to verbascoside.
Line 270: what relative remaining activity was performed here?
Answer: This change had been made accordingly- see comments made
If this is a different method, authors should sectionalized the various methods to make the methodology readable as in its current state it is confusing
Answer: These changes have been made accordingly
4.4 DPPH and NO......
Answer: This change had been made accordingly- see comments made
What was the final concentration of EtOH in the reaction as this has effects on the DPPH assay.
Answer: 10% EtOH in final solution
How was the 10% EtOH concentration chosen as blank?
Answer: As the concentration of the final solution contained 10% EtOH, a blank with 10% EtOH was used to indicate no effect by the solvent used
Line 289: Define colour or negative controls
Answer: The colour of control or negative controls indicate the full reaction without the addition of DPPH. As herbal extracts when testing for DPPH inhibition has a characteristic green colour which may overshadow the pink colour of DPPH or the yellow colour shown when inhibition takes place. This is used so that the green colour of the herbal extract may be subtracted when doing analysis.
Line 290: Check the word Absorbencies????
Answer: Absorbances
Line 293: Check the word Absorbencies??
Answer: Absorbances
Line 299: Same as above
Line 302: same as above
Line 331: Sentence is wrong. Kindly rephrase
Answer: This change had been made accordingly
What solvent was used in this session?
Answer: DMSO was used with a final concentration of 2% in the final solution
Concentrations should be from 1.53-400 not 400- to 1.53ug/mL. range should be from low to high unless units are not the same
Answer: This change had been made accordingly- see comments made
Same as above
Answer: This change had been made accordingly- see comments made

Reviewer 3 Report
In this manuscript the authors explore in vitro metabolism, antioxidant and cytotoxic properties of verbascoside. The authors do a very good job on this project including design, data collection and interpretation. Below are some comments that would clarify and strengthen the manuscript.
I could not access the supplemental files so I don't know if it shown there but including a structure would be helpful.
In the introduction lines 36-37 the authors mention "many publications...." yet they only cite two, refs 4 and 5. "many" is generally much more than 2.
Table 1 has a header as "Structure" yet only a formula is listed, suggest changing the heading to "Formula"
In table 2 for CYP2C19 with substrate TFD032 the data "Stimulation" is listed. It is footnoted but the footnote does not adequately explain what the term means. Please explain.
In section 2.4 the authors present the results of the cellular proliferation assay and mentioned the compound having weak antiproliferative effect based on IC50 values > 100 micromolar. What is the basis for 100 micromolar in classifying it as a weak antiproliferative? The authors should expand this section.
Author Response
I could not access the supplemental files so I don't know if it shown there but including a structure would be helpful.
Answer: We’re sorry about that, therefore, we have transformed the files as PDF format
In the introduction lines 36-37 the authors mention "many publications...." yet they only cite two, refs 4 and 5. "many" is generally much more than 2.
Answer: This change had been made accordingly- see comments made
Table 1 has a header as "Structure" yet only a formula is listed, suggest changing the heading to "Formula"
Answer: This change had been made accordingly- see comments made
In table 2 for CYP2C19 with substrate TFD032 the data "Stimulation" is listed. It is footnoted but the footnote does not adequately explain what the term means. Please explain.
Answer: “Stimulation” is not “induction” mediated by increase in mRNA, stimulation can be explained, for instance, due to “an allosteric” binding to the enzyme. The explanation of the stimulation was added to the footnote.
In section 2.4 the authors present the results of the cellular proliferation assay and mentioned the compound having weak antiproliferative effect based on IC50 values > 100 micromolar. What is the basis for 100 micromolar in classifying it as a weak antiproliferative? The authors should expand this section.
Answer: This change had been made accordingly- see comments made

Round 2
Reviewer 1 Report
The authors did their very best and addressed all comments.